# Financing Constraints, Carbon Emissions and High-Quality Urban Development—Empirical Evidence from 290 Cities in China

**DOI:** 10.3390/ijerph19042386

**Published:** 2022-02-18

**Authors:** Shaobo Wang, Junfeng Liu, Xionghe Qin

**Affiliations:** 1Institute of County Economic Development, Lanzhou University, Lanzhou 730000, China; wshb@lzu.edu.cn; 2School of Business, Suzhou University of Science and Technology, Suzhou 215009, China; 3Institute for Global Innovation and Development, School of Urban and Regional Science, East China Normal University, Shanghai 200050, China

**Keywords:** financing constraints, carbon emissions, high-quality urban development, intermediary effect

## Abstract

The tightening of the financing environment and global climate change have become urgent problems for high-quality economic development all over the world. Facing these challenges, the Chinese government is committed to alleviating regional financing constraints and setting carbon-emission reduction targets. However, are these measures effective for high-quality urban development? This paper attempts to use unbalanced panel data from 290 cities on the Chinese mainland from 2004–2017 to provide an answer to the problem using a scatter plot and the mediator effect model. Results show that: (1) financing constraints limit the funds required for urban development, which is not conducive to high-quality urban development, but high-quality urban development has the characteristics of “path dependence”; (2) In the context of environmental regulation, financing constraints are mainly enacted through reducing carbon emissions, which is inconducive to high-quality urban development. Carbon emissions are the transmission mechanism whereby financing constraints affect high-quality urban development; (3) Cities with large financing constraints have insufficient capital investment for high-quality urban development, and the aggravation of financing constraints has an increasingly obvious inhibitory effect on high-quality urban development. Moreover, due to the effect of the global economic crisis in 2008, the negative effect of financing constraints on high-quality urban development had the characteristics of U-shaped fluctuation. Thus, this paper believes that the implementation of China’s double carbon policy is at the expense of high-quality urban development, and there is a long way to go before high-quality urban development reaches later stages. Other countries should carefully weigh up the relationship between environmental pollution and economic development when facing financing constraints.

## 1. Introduction

Despite increasing social material wealth, the industrial revolution has also caused irreversible damage to Earth’s environment. Global climate change caused by carbon emissions has also become the focus of attention all over the world [1]. The academic community has increasingly deepened research on the relationship between economic development and carbon emissions [2]. At the same time, with the background of global economic weakness, the social financing environment has become another leading factor affecting economic development. Since the turn of the 21st century, China’s economic development and carbon emissions have shown an obvious upward trend [3,4], and the financing environment has gradually improved with the background of marketization. Therefore, China is the research object of this paper. Although in his report to the 19th National Congress of the CPC National Congress, General Secretary Xi Jinping clearly stated that “China’s economy has shifted from a stage of rapid growth to a stage of high-quality development”, affected by financial system reform and dual-carbon policy objectives, the financing environment [5] and carbon emission issues [6] present a new challenge for high-quality urban development. Therefore, against the background of China’s financial development and environmental requirements, and with the goal of promoting high-quality urban development based on non-balanced panel data from 290 prefecture-level cities in China, this paper discusses the relationship between financing constraints, carbon emissions, and high-quality urban development, trying to realize high-quality urban development in China and provide a Chinese solution for the implementation of financial reform and carbon-emission reduction policies in other countries.

The existing literature has studied the relationship between financing constraints, carbon emissions, and high-quality urban development from different angles, mainly including the following three aspects: (1) Regarding the effect of financing constraints on high-quality development, scholars have had a heated discussion. From the perspective of green finance, some studies believe that green finance is conducive to sustainable economic development [7,8,9]. Most existing scholars measure high-quality development from total factor productivity, and it is generally believed that financing constraints have inhibitory effects on total factor productivity [10]. Some scholars have objected that the improvement of financing constraints does not improve high-quality development [11], and that financing constraints even aid high-quality development [12,13]; (2) Regarding the effect of financing constraints on carbon emissions, most scholars believe that financing constraints may limit the intensity of regional capital investment and are inconducive to green technology innovation, which will increase carbon emissions [14,15]. Few scholars believe that moderate financing constraints will force managers to concentrate on superior capital for technological innovation [16,17], therefore easing carbon emissions; (3) Regarding the effect of carbon emissions on high-quality urban development, the premise of carbon emissions is energy consumption. Some scholars believe that energy consumption can effectively promote high-quality urban development [18,19]. Opponents believe that energy consumption is inconducive to high-quality urban development [20,21]. In addition, a few scholars believe that the effect of energy consumption on high-quality urban development is insignificant [22,23].

Based on the above literature, the existing research on the relationship between financing constraints, carbon emissions, and high-quality urban development has achieved rich results in the empirical field, but the literature on financial reform and carbon-emission reduction given requirements of high-quality urban development under the unified analysis framework is relatively rare. Moreover, there are still great differences in the empirical results of the relationships between the three. Therefore, compared with the existing literature, the contribution of this paper is as follows: (1) To clarify the theoretical mechanism between financing constraints, carbon emissions, and high-quality urban development. Based on the existing research, this paper discusses the influence mechanism of financing constraints on high-quality urban development and financing constraints on high-quality urban development through carbon emissions; (2) The implementation of China’s dual-carbon target policy is at the expense of high-quality urban development, and China has acted as a major country for world ecological and environmental protection. At the same time, this provides a reference for other countries to develop their economies. We should grasp the relationship between carbon emissions and high-quality urban development, seek the best point of interaction, and minimize carbon emissions while being committed to high-quality development; (3) To open the “black box” that shows how financing constraints affects the high-quality development of cities, i.e., through carbon emissions. Empirical results show that financing constraints can affect high-quality urban development through carbon emissions.

The following contents are arranged as follows: the next section is mechanism analysis, which explains in detail the impact of financing constraints on high-quality urban development, and the effect of financing constraints on high-quality urban development through carbon emissions. The third section is the study design, which introduces the measurement model, variable measurement, and data description, and shows the possible relationship between financing constraints and high-quality urban development in the form of a scatter diagram. The fourth section is the empirical test results, which examines whether financing constraints really inhibit the high-quality development of cities, discusses the intermediary effect based on carbon emissions, and tests heterogeneity at the same time. The fifth section is the discussion. Based on research results, it presents countermeasures and suggestions for China’s future financial system reform, the system of carbon-emission reduction policy, and the path of high-quality urban development. At the same time, it provides China’s experience for countries around the world to weigh up the trade-off between environmental pollution and economic development. The sixth section is the conclusion, which summarizes the full text, shows the core conclusions of this paper, and points out research limitations and prospects of this paper.

## 2. Mechanism Analysis

### 2.1. The Effect of Financing Constraints on the High-Quality Urban Development

There is an inhibitory mechanism of financing constraints on high-quality urban development. Most of the existing literature believes that financing constraints caused by “difficult and expensive financing” have a negative effect on high-quality urban development [24]. Financing constraints caused by credit distortion make overall regional financing costs excessively high and reduce sufficient financial support, which makes it difficult to ensure funds are available for regional development. As a result, resource allocation and economic development potential in a region cannot reach the optimal level, which will then affect the high-quality development of a city. The negative effect of financing constraints on high-quality urban development is mainly reflected by the following two aspects: (1) Financing constraints hinder the optimal allocation of resources [25]. This mainly refers to occasions when a region suffers from financing constraints, and the rise of external financing costs increases and restricts the investment portfolio of production factors, such as fixed-asset investment, R&D expenditure, capital investment, and labor investment in cities, which are inconducive to high-quality urban development. (2) Financing constraints may reduce the capital conditions required for high-quality urban development regarding economic growth [26]. This mainly refers to the fact that with financing constraints, the production and operation of enterprises in cities cannot guarantee the funds required for human capital expenditures, operating expenditures, and other business expenditures. This causes the development of urban industries to lag behind, restricts economic growth, and affects the high-quality development of cities.

Financing constraints plays a role in promoting high-quality urban development. Most existing studies believe that a reverse relationship exists, but some scholars believe that although financing constraints are inconducive to regional financing, they may promote high-quality urban development through other factors. This view is based on the incentive effect of financing constraints, i.e., financing constraints can encourage a regional economy to make breakthroughs in innovation and efficiency, thus promoting high-quality urban development: (1) Although financing constraints inhibit capital source channels of regional economic development, they can stimulate the competitive potential of the regional economy. To survive and strengthen innovation capacity, enterprises in the region must concentrate limited funds on overcoming technological difficulties. Through R&D and innovation, this can promote the high-quality development of an enterprise and realize the high-quality development of the city [27]. (2) Under the effect of financing constraints, a regional economy cannot guarantee the financial conditions required for economic growth. Enterprises in a region will optimize internal management and intensive production to increase the use rate of funds, and, in the short term, improve the level of high-quality urban development using non-financing means [28].

### 2.2. The Effect of Financing Constraints on High-Quality Urban Development through Carbon Emissions

The effect of financing constraints through carbon emissions on high-quality urban development is uncertain. This uncertainty is mainly reflected in the influence path of the two phases:

In the first stage, the effect of financing constraints on carbon emissions is uncertain: (1) Financing constraints have aggravating effects on carbon emissions. Due to the large uncertainty of green technology innovation, capital investment must be long-term and sustainable [29], and insufficient funds caused by financing constraints may lead to R&D interruption and the loss of technical personnel, which is inconducive to green technology innovation [30], and makes carbon emissions high. In addition, environmental protection investment is a long-term investment with low short-term return. Financing constraints may force local governments to make limited investments in environmental protection and low investment willingness [31,32], which will lead to increased carbon dioxide emissions; (2) Financing constraints may also have a mitigating effect on carbon emissions. In the context of strong regional financing constraints, establishing high-polluting, high-energy-consuming, and high-emission industrial enterprises is impossible, meaning regions can only have a light industrial production system, which has a significant mitigation effect on carbon emissions. Sadorsky (2010) and Zhang (2011) discovered, from a financial perspective, that financial development will increase carbon emissions, i.e., financing constraints may alleviate carbon emissions [33,34].

In the second stage, the effect of carbon emissions for the high-quality development of cities is uncertain: (1) On the one hand, carbon emissions contribute to the high-quality development of cities. The increase in carbon emissions is mostly based on resource consumption. Resource consumption is not only an important source of economic growth, but it also means an increase in the level of industrialization and the development of productivity. Therefore, carbon emissions contribute to the high-quality development of cities [21]; (2) On the other hand, an increase in carbon emissions inhibits the high-quality development of cities. Some scholars believe that the increase of carbon emissions aggravates environmental pollution problems such as haze, reduces the urban attraction and urbanization process [35,36], and destroys the accumulation of human capital [37,38]; (3) The EKC curve shows that an inverted U-shaped relationship exists between the severity of environmental pollution and the growth rate of the economy, i.e., rapid growth of economic aggregate is promoted at the expense of environmental damage in the early stages of production, while the reduction of resources and low productivity in later stages inhibits economic development [39].

Based on this comprehensive literature and theoretical mechanism analysis, we can judge as follows: (1) Financing constraints are an important factor affecting high-quality urban development, but debates on whether the final results are suppressed or promoted remain. (2) The effect of financing constraints on carbon emissions and carbon emissions on high-quality urban development is uncertain, and the effect of financing constraints on high-quality urban development through carbon emissions is uncertain. Therefore, this paper tries to include “financing constraints, carbon emission, and high-quality urban development” into a unified analysis framework using the Chinese mainland’s 290 prefecture-level cities as the research sample to investigate the direct effect of financing constraints on high-quality urban development, explore the role of carbon emissions in the process of financing constraints affecting the high-quality development of cities, and provide a policy basis for the alleviation of financing constraints, the achievement of dual-carbon goals, and the path to high-quality urban development in the next stage.

## 3. Study Design

### 3.1. Measurement Model Setting

Based on the literature review and the theoretical mechanism analysis, financing constraints can affect the high-quality development of the cities, but the direction, degree, and transmission mechanism of the financing constraints affecting high-quality urban development needs to be analyzed through empirical tests. Therefore, based on the intermediary effect test method constructed by Baron & Kenny (1986) and Wen et al. (2014) [40,41], combined with the bootstrap method proposed by Precher & Hayes (2004), this paper constructs the following recursive model to test the effect of financing constraints on high-quality urban development and the intermediary effect of carbon emissions [42].
(1)$$hq_{it}=\alpha_{0}+\theta_{1}fc_{it}+\sum \beta_{i}control_{it}+city_{i}+year_{t}+\varepsilon_{it}$$
(2)$$co2_{it}=\alpha_{0}+\varphi_{1}fc_{it}+\sum \beta_{i}control_{it}+city_{i}+year_{t}+\varepsilon_{it}$$
(3)$$hq_{it}=\alpha_{0}+\alpha_{1}fc_{it}+\alpha_{2}co2_{it}+\sum \beta_{i}control_{it}+city_{i}+year_{t}+\varepsilon_{it}$$

In the measurement model, *i* represents the individual city, *t* represents the year, *hq* represents the high-quality development level of the city, *fc* represents the financing constraint, *co*2 is the intermediary variable carbon emissions, *control* is the control variable, *city* is the unobservable individual fixed effect of the city, *year* is the time-fixed effect and $\varepsilon_{it}$ is the random disturbance item. With reference to Petersen (2009) [43], when a two-way fixed effects regression is conducted in this paper, the region is “cluster” processing, while the robust standard error of the coefficient estimate is generated. Among them, model (1) mainly examines the effect of financing constraints on the high-quality development of a city, and comprehensive models (1), (2), and (3) test whether carbon emissions (*co*2) are an intermediary variable for financing constraints affecting high-quality development.

The test steps of intermediary effect are as follows: the first step is to estimate model (1) and observe whether the regression coefficient $\theta_{1}$ of financing constraints (*fc*) on high-quality urban development (*hq*) is significant. If the coefficient $\theta_{1}$ is significant, it indicates that the financing constraint can affect the high-quality development of cities, and can be further tested. If the coefficient $\theta_{1}$ is not significant, it indicates that the intermediary effect is not tenable, and there is no need for subsequent regression. The second step is to estimate based on model (2) and test the significance of the regression coefficient $\varphi_{1}$ between financing constraints (*fc*) and carbon emissions (*co*2). Thirdly, based on the estimation results of model (3), if the coefficients $\alpha_{1}$ and coefficients $\alpha_{2}$ have a certain significance level, and $\alpha_{1}$ is reduced compared with $\theta_{1}$, it indicates that there is some intermediary effect. If the coefficient $\alpha_{1}$ is not significant, and the coefficient $\alpha_{2}$ is significant, it indicates that carbon emission (*co*2) plays a complete intermediary role in the impact of financing constraints (*fc*) on high-quality urban development (*hq*).

### 3.2. Variable Measurement and Data Description

The explained variable is the high-quality urban development (*hq*). Since the formulation of high-quality development is relatively new, there is no consensus on the definition and measurement of high-quality development in the existing literature [44,45]. The existing methods of high-quality evaluation can be divided into two types: the first builds an index system to measure high-quality urban development level from multiple perspectives; the second measures urban high-quality development level with a single index, such as total factor productivity (*TFP*). This paper believes that the core of high-quality urban development lies in production efficiency and development potential. The key to high-quality urban development lies in whether regional total factor productivity can be steadily improved. Therefore, although the concept of total factor productivity cannot fully include the connotation of high-quality urban development, it has become the most popular index to measure China’s high-quality economic development [46,47]. This paper uses urban TFP to characterize high-quality urban development levels.

The explanatory variable is the regional financing constraint (*fc*). Financing constraints are generally based on micro-enterprise-level research and may refer to the level of external financing. There are few existing measures of regional financing constraints. Therefore, referring to the study [48,49], regional financial inhibition indicators are used as a proxy variable for regional financing constraints. According to the study [50], first, the degree of financial development is calculated, i.e., the balance of RMB loans of financial institutions at the end of the year/GDP; second, the use efficiency of bank funds is calculated, i.e., the balance of various RMB deposits of financial institutions at the end of the year, the balance of various RMB loans of financial institutions at the end of the year, and the balance of various RMB deposits of financial institutions at the end of the year. A weighted average of financial development degree and bank fund use efficiency can form financial inhibition indicators to characterize financing constraints.

The mediation variable is carbon emission (*co*2). Referring to Lin & Liu (2010), the carbon emissions mentioned in this paper refer to CO_2_ emissions related to energy activity, wherein CO_2_ mainly arises from the combustion of coal, fossil oil, and gas [51]. The specific calculation formula is: $C_{i}=\sum \alpha_{i}\beta_{i}E_{i}$, where *C_i_* is the CO_2_ emission of the i-th energy source, *α_i_* is the conversion rate of such an energy source, and *β_i_* is its CO_2_ emission coefficient. *E_i_* is the consumption of coal, fossil oil, and gas, which is equivalent to total primary energy consumption multiplied by the respective proportion of the primary energy consumption.

In terms of control variables, we refer to the relevant literature based on data availability: (1) economic development level (*gdp*), which is measured by the logarithm value of actual GDP; (2) the degree of openness (*fdi*), which is measured by the proportion of actual foreign investment in GDP; (3) industrial structure (*sec*), which is measured by the proportion of the added value of the secondary industry in GDP; (4) infrastructure construction (*infra*), which is measured by the proportion of total passenger traffic in the total population at the end of the year; (5) population agglomeration (*popu*), which is measured by the ratio of total population to administrative area at the end of the year; (6) informationization level (*infor*), which is measured by the proportion of post and telecommunication business income in GDP; and (7) fixed-asset investment (*cap*), which is measured by the logarithm of the total fixed-asset investment.

Descriptions of the relevant variables where financing affects high-quality urban development are given in Table 1.

Considering the data availability of some indicators, this paper planned to select 290 prefecture-level cities in China from 2004 to 2017 as the research areas, and the relevant data are derived from the China Urban Yearbook and China Energy Statistics Yearbook. To avoid regression errors due to outliers, a 1% tail reduction of related continuous variables was carried out, followed by the Winsorization method. Table 2 presents the descriptive statistics of the variables related to financing that affect high-quality urban development.

### 3.3. Analysis of the Basic Facts

To clarify the possible relationship between financing constraints on high-quality urban development and carbon emissions, this paper uses a scatter diagram to show the relationship between financing constraints and high-quality urban development (Figure 1) and the relationship between financing constraints and carbon emissions (Figure 2). According to the figure analysis, financing constraints show a negative correlation with high-quality urban development and carbon emissions. It is preliminarily believed that financing constraints are inconducive to high-quality urban development and carbon emissions, i.e., financing constraints may hinder the improvement in quality of cities, and financing constraints are conducive to reducing carbon emissions. The series of questions in this article is; (1) is this the case? (2) what role do carbon emissions play between financing constraints and high-quality urban development? and (3) What is the impact of heterogeneous financing constraints on high-quality urban development? Further investigation using regression equations is necessary.

## 4. Test of Financing Constraints, Carbon Emissions, and High-Quality Urban Development

### 4.1. Direct Effect of Financing Constraints on High-Quality Urban Development

Table 3 shows the regression results of financing constraints affecting high-quality urban development. Among them, model (1) is a mixed regression result, and model (2) is a two-way fixed regression result. In addition, to explore the possible path-dependence characteristics of a city’s high-quality development, a dynamic panel model was set. Model (3) is the system GMM estimation result, while model (4) is the differential GMM estimation result. Results show that: (1) the results of the mixed-panel estimation and the two-way fixed effects model showed that financing constraints have a significant inhibitory effect on the high-quality development of cities, i.e., financing constraints are inconducive to high-quality urban development; (2) the results of the systematic *GMM* and differential *GMM* confirmed that financing constraints are inconducive to the high-quality development of an enterprise. The coefficient of the first-order lag of high-quality urban development is positive at the 1% significance level, indicating high-quality urban development. It has certain “path-dependence characteristics”, i.e., the high-quality development level of a city in the early stages plays an important role in promoting the high-quality development of the current city; and (3) in terms of control variables, the level of economic development (*gdp*) and fixed-asset investment (*cap*) are significantly negative. The analysis suggests that China’s economic growth in early stages mainly relied on investment, and the pursuit of growth speed ignored growth quality. Therefore, the increase in GDP and the increase in fixed-asset investment level are inconducive to high-quality urban development; the degree of openness (*fdi*), industrial structure (*sec*), infrastructure (*infra*), and population agglomeration (*popu*) are significantly positive. The analysis suggests that the degree-of-openness factors, such as the increase in the number of cities, the optimization and upgrade of the industrial structure, the expansion of infrastructure construction, and the increase in the degree of population agglomeration, can effectively promote the high-quality development of the city.

### 4.2. Mediation Effect Test of Financing Constraints Affect the High-Quality Development of Cities through Carbon Emissions

The direct effect test results show that financing constraints may significantly hinder the high-quality development of cities. However, through which channels do financing constraints affect the high-quality development of cities? Theoretical analysis shows that carbon emissions play an important role in the process of financing constraints affecting the high-quality development of cities. Therefore, this section will examine the mediating effect of financing constraints on the high-quality development of cities through carbon emissions. Models (1) to (3) and (4) to (6) in Table 4 report the mixed-panel and two-way fixed effect estimation results. According to the analysis in Table 4: (1) financing constraints have effectively reduced carbon emissions. The analysis suggests that under the background of financing constraints, enterprises are gradually progressing to high-quality production methods, coupled with the closure of some high-energy, high-pollution, and high-emission enterprises, in which carbon emissions are showing a downward trend; (2) carbon dioxide emissions contribute to the high-quality development of cities. The analysis suggests that although carbon emissions have an adverse effect on the environment and can also reflect the level of resource consumption, they can effectively contribute to the high-quality development of cities; and (3) Although the two-way fixed effect estimation results show that carbon dioxide emissions are insignificant for the high-quality development of enterprises, after bootstrap testing, test results show that carbon emissions are the intermediary variables of FDI inflows affecting high-quality economic development, i.e., financing constraints are reduced by reducing carbon dioxide emissions, which are inconducive to the high-quality development of cities.

### 4.3. Test of the Heterogeneity of Financing Constraints Affecting High-Quality Urban Development

To further examine the heterogeneity effect of financing constraints on high-quality urban development, this paper will explore the effect of different quantile financing constraints on high-quality urban development and the effect of different stage financing constraints on high-quality urban development.

#### 4.3.1. Heterogeneity Effect of Different Quantile Financing Constraints on High-Quality Urban Development

This paper divides the full sample into five parts according to the 10th, 25th, 50th, 75th, and 90th quantiles of financing constraints, and performs a group regression. The regression results are shown in Table 5. According to the table: (1) the regression results of each quantile show that financing constraints have an inhibitory effect on the high-quality development of cities, indicating that the conclusion that financing constraints are inconducive to the high-quality development of cities has a certain degree of robustness; and (2) comparing the changes in the financing constraint coefficients of the regression results, as the degree of financing constraints increases, the adverse effects of financing constraints on the high-quality development of cities are becoming greater. Analysis suggests that cities with greater financing constraints have insufficient capital investment for high-quality urban development, while their ability to attract foreign funds is weak. Under this dual background, the elements required for high-quality urban development are lacking, and cannot support high-quality urban development. The negative effect on the high-quality development of a city becomes obvious.

#### 4.3.2. Heterogeneity Effect of Financing Constraints at Different Stages on High-Quality Urban Development

To explore the time trend of financing constraints affecting the high-quality development of cities, this paper conducted a group regression according to four time nodes of 2006, 2010, 2014, and 2017. The regression results are shown in Table 6, models (1) to (4). According to the table: (1) financing constraints have not been conducive to the high-quality development of Chinese cities in recent years, indicating that China’s financing environment needs to be further improved; and (2) the regression coefficient in 2010 was significantly higher than that in 2006, indicating that the global economic crisis in 2008 had a strong effect on China’s economic and financial sectors; the regression coefficient in 2014 was the same as in 2010. This shows that the adverse effects of the economic crisis have been continuing, while China was undergoing a short adjustment stage from high-speed growth to high-quality growth at that time, making the negative effect of financing constraints on the high-quality urban development more obvious; the regression coefficient in 2017 was obviously lower than in 2014 and basically the same as in 2006, indicating that after a short period of adjustment in China, the negative effect of financing constraints on the high-quality development of cities has returned to normal. It is foreseeable that financing constraints will affect the high-quality development of cities. The negative effect will gradually decrease.

## 5. Discussion

This article used financial reform, dual-carbon policy goals, and the high-quality development of the Chinese economy as a background and closely focused on the relationship between financing constraints, carbon emissions, and high-quality urban development, thus clarifying the basis of the relationships between the three. Using unbalanced panel data of 290 prefecture-level cities in mainland China from 2004 to 2017 as the research sample, the mediation effect model is used to test the effect of financing constraints on the high-quality development of cities, and the role of carbon emissions between the two. Specific results are as follows:

First, financing constraints are inconducive to high-quality urban development. This conclusion is consistent with most existing studies [52]. According to the actual situation of China’s economy, as a developing country, China’s financial system is not perfect. It is normal for financing constraints under financial repression to have an adverse impact on high-quality urban development [53], mainly because financing constraints limit the capital elements required for urban development and seriously affect the efficiency of factor allocation. This leads to the lack of internal driving force for high-quality urban development. Therefore, under the premise of ensuring systemic financial risks do not occur, all countries in the world should gradually reduce financing constraints. Furthermore, banks and other financial institutions should appropriately relax credit standards and innovate the form of collateral with the aim of injecting more capital vitality into urban development; as enterprises, they can expand their funding sources through mutual guarantees, therefore alleviating financing constraints and promoting the high-quality development of enterprises and cities; as a government, they may provide key financial support to sunrise industries, such as technological innovation, improve the efficiency of capital use, and promote high-quality urban development through intensive forms.

Second, financing constraints affect the high-quality development of cities by influencing carbon emissions, i.e., carbon emissions are the transmission mechanism between financing constraints and high-quality urban development. The mediation effect test results show that financing constraints effectively suppress carbon emissions, but carbon emissions contribute to the high-quality development of cities. This also verified the conclusion that financing constraints are inconducive to the high-quality development of cities by suppressing carbon emissions. The analysis shows that when cities face high financing constraints, a government has insufficient investment in environmental governance, and enterprises cannot effectively carry out green technology innovation [30,31]. High financing constraints lead to an increase in carbon emissions. Carbon emission is an important factor affecting high-quality urban development [54]. Therefore, carbon emission is an intermediary variable of financing constraints affecting high-quality urban development. According to the above conclusion, the current increase in China’s carbon emissions is necessary for the improvement of high-quality urban development. However, with the background of dual-carbon goals, China’s carbon-emission reduction plan is a foregone conclusion, which means that the high-quality development of Chinese cities will face carbon-emission constraints. Furthermore, to protect the ecological environment, China has sacrificed economic growth and high-quality urban development. For countries around the world, financing constraints are one of the important factors affecting carbon emissions and high-quality urban development, and carbon emissions contribute to economic growth and high-quality urban development [55]. Countries need to strike a balance between environmental protection and high-quality urban development.

Third, the higher the level of urban financing constraints, the less conducive it is to the high-quality development of cities. In addition, financing constraints have had an obvious negative effect on the high-quality development of cities. The heterogeneity test results show that, on the one hand, different quantile financing constraints on high-quality urban development show that various degrees of financing constraints have an inhibitory effect on high-quality urban development. Simultaneously, with a higher the degree of financing constraints, the adverse effects on high-quality urban development become more obvious. On the other hand, from 2004 to 2017, the inhibitory effect of financing constraints on high-quality urban development has shown a trend of strengthening first and then alleviating. In the post-economic-crisis period, the restraining effect of financing constraints on the high-quality development of cities is more pronounced. The analysis shows that with the background of high financing constraints, the level of financial development is low, which has a negative impact on economic development and high-quality urban development [49]. For cities with low financing constraints, a sound financial system can support economic development, so financing constraints have little impact on high-quality urban development. In addition, the heterogeneous impact of financing constraints at different stages on high-quality urban development shows that, since China’s entry into WTO, the characteristics of the market economy have become increasingly obvious, and financing constraints have been gradually alleviated [56]. Simultaneously, the negative impact of financing constraints on high-quality urban development has also been alleviated. Combined with the development process of globalization, this paper believes that when alleviating regional financing constraints and promoting high-quality urban development, all countries should fully consider the differences of urban financing constraints, and the implementation of targeted mitigation strategies for areas with high financing constraints is more conducive to high-quality urban development. Despite alleviating financing constraints, governments of all countries can achieve a balance of high-quality urban development. Countries around the world can gradually expand their opening-up to the outside world, which will help alleviate financing constraints and achieve high-quality urban development [57,58].

## 6. Conclusions

Existing studies on the effect of financing constraints on high-quality development offer two conclusions, positive and negative. However, as far as the conclusions of this article are concerned, financing constraints at the regional level are inconducive to high-quality urban development. This paper also finds that an increase in carbon emissions is based on the premise of resource consumption. Although resource consumption may cause environmental pollution and ecological damage, resource consumption also provides momentum for the high-quality development of cities, and carbon emissions help high-quality urban development. At the same time, results show that high-quality urban development is the result of multiple factors, such as financing constraints and carbon emissions. Visible countermeasures should reduce financing constraints and increase carbon emissions, aiming to promote high-quality urban development. However, due to the effect of financial reforms and dual-carbon targets, there are major obstacles to the reduction of financing constraints and the increase of carbon emissions. Therefore, for all countries in the world, it is necessary to alleviate financing constraints and promote high-quality urban development, but it is difficult to achieve environmental protection and high-quality urban development at the same time. Governments of all countries should accelerate the reform of the financial system, and focus on reasonable carbon emissions, thus developing advantageous industries and promoting high-quality urban development from multiple perspectives, using policy support and element guidance.

It is undeniable that the research of this paper has limitations. The existing measures of high-quality urban development are diverse. Therefore, the use of TFP to measure high-quality urban development in this paper may be biased or one-sided. Furthermore, due to the insufficient availability and effectiveness of data, other variables are not controlled, and there may be a problem of missing variables. Therefore, in future research, we will consider the connotation of high-quality urban development as much as possible, add more control variables when the data are available, and then accurately explore the relationship between financing constraints, carbon emissions, and high-quality urban development.

## Figures and Tables

**Figure 1 ijerph-19-02386-f001:**
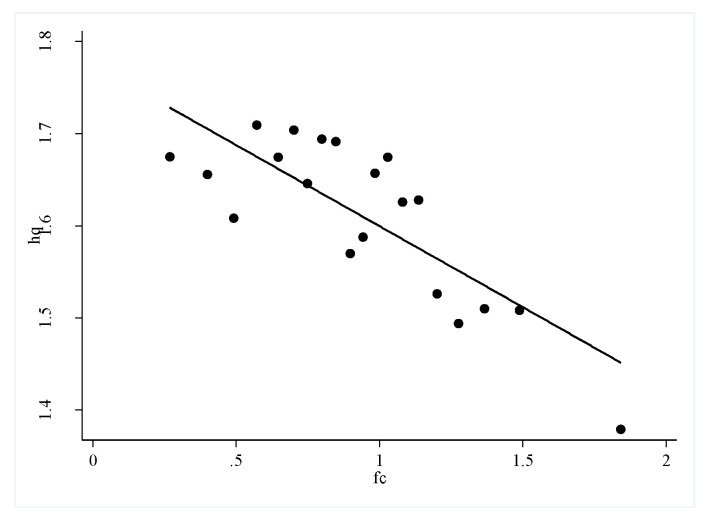
Scatter fit between financing constraints.

**Figure 2 ijerph-19-02386-f002:**
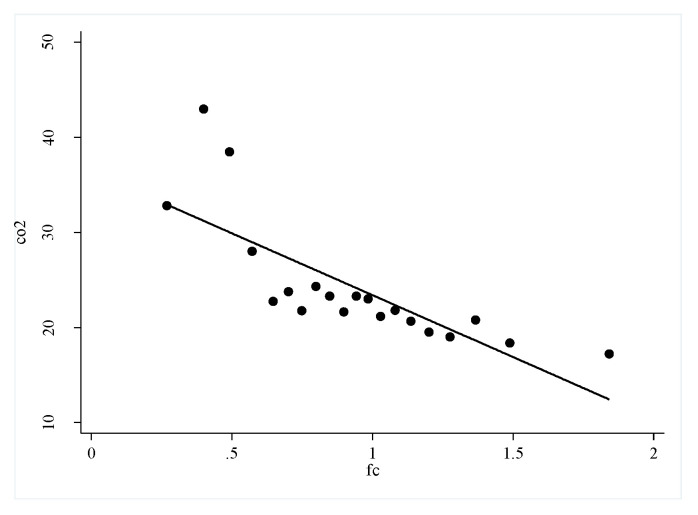
Scatter fit between financing constraints and high-quality urban development carbon emissions.

**Table 1 ijerph-19-02386-t001:** Variables description.

Variables	Definitions	Symbols	Measurement Methods
Dependent Variable	High-quality urban development	*hq*	The logarithm value of total factor productivity measured by OLS
Independent Variable	financing constraints	*fc*	financing constraints was used as the proxy variable
Mediator Variable	carbon emissions	*co*2	*C_i_* = Σ*α_i_β_i_E_i_*
Control Variables	economic development level	*gdp*	The logarithm value of GDP
	The degree of openness	*fdi*	The proportion of the actual foreign investment in GDP
	Industrial structure	*sec*	The added value of the secondary industry accounts for GDP
	Infrastructure construction	*infra*	The proportion of total passenger traffic in the total population at the end of the year
	Population agglomeration	*popu*	The ratio of the total population to the administrative area at the end of the year
	Information level	*infor*	The proportion of post and telecommunications business income in GDP
	Fixed-asset investment	*cap*	The logarithm of the total fixed-asset investment
	Time effect	*year*	Virtual variable of the year
	Regional effect	*city*	Virtual variable of the region

**Table 2 ijerph-19-02386-t002:** Descriptive statistics.

Variables	*hq*	*fc*	*co*2	*gdp*	*fdi*	*sec*	*infra*	*popu*	*infor*	*cap*
Mean	1.596	0.946	24.9	10.17	0.0204	48.84	22.22	−3.484	0.0278	15.48
Std. deviation	0.786	0.382	20.28	0.773	0.021	10.78	22.57	0.871	0.0165	1.137
Minimum	0.108	0.198	1.672	8.039	0.000119	21	3.052	−6.012	0.00658	12.27
Maximum	2.917	2.141	103.4	11.81	0.15	78.7	144	−2.006	0.0975	17.85
The 10th quantile	0.468	0.454	6.434	9.133	0.00196	35.5	7.318	−4.756	0.0128	13.93
The 25th quantile	0.928	0.683	10.55	9.629	0.00573	42	10.62	−4.006	0.0172	14.66
The 50th quantile	1.666	0.932	18.63	10.21	0.0132	49.2	15.53	−3.357	0.0235	15.55
The 75th quantile	2.266	1.183	33.03	10.73	0.0279	55.6	23.97	−2.771	0.0329	16.29
The 90th quantile	2.611	1.434	53.91	11.16	0.0481	61.8	41.86	−2.5	0.048	16.93
Observations	4060	3967	4060	3935	3498	3718	3144	4011	3943	3719

**Table 3 ijerph-19-02386-t003:** The direct impact of financing constraints on urban high-quality development.

Model	(1)	(2)	(3)	(4)
Method	POLS	FE	SYS-GMM	DIF-GMM
Variable	*hq*	*hq*	*hq*	*hq*
*L.hq*			0.3804 ***	0.4434 ***
			(0.0013)	(0.0006)
*fc*	−0.2066 ***	−0.4761 ***	−1.1918 ***	−0.8428 ***
	(0.0360)	(0.0583)	(0.0151)	(0.0083)
*gdp*	−0.2311 ***	−0.1472 *	−0.5490 ***	−0.4216 ***
	(0.0344)	(0.0809)	(0.0175)	(0.0092)
*fdi*	2.8507 ***	0.7007	2.0302 ***	−1.7307 ***
	(0.6142)	(0.8038)	(0.2228)	(0.2057)
*sec*	0.0070 ***	0.0199 ***	0.0581 ***	0.0289 ***
	(0.0015)	(0.0028)	(0.0011)	(0.0005)
*infra*	0.0027 ***	0.0059 ***	0.0060 ***	0.0029 ***
	(0.0007)	(0.0011)	(0.0002)	(0.0002)
*popu*	0.0400 **	−0.6053 **	0.1281	0.3311 ***
	(0.0182)	(0.2973)	(0.1312)	(0.0068)
*infor*	−0.9138	−1.6706	−1.1627 ***	−8.9904 ***
	(0.8607)	(1.1349)	(0.3449)	(0.1393)
*cap*	−0.1386 ***	−0.3603 ***	−0.3198 ***	−0.2579 ***
	(0.0208)	(0.0525)	(0.0095)	(0.0066)
*_cons*	5.7464 ***	5.6976 ***	9.7953 ***	9.6649 ***
	(0.2512)	(1.1239)	(0.5034)	(0.0402)
*year*	NO	YES	YES	YES
*city*	NO	YES	YES	YES
*N*	2993	2989	2397	2993
*R−sq*	0.127	0.221		

Notes: Clustered standard errors are reported in (). Significant at * 10%, ** 5%, and *** 1%.

**Table 4 ijerph-19-02386-t004:** The impact of financing constraints on high-quality urban development through carbon emissions.

Method	POLS			FE		
Model	(1)	(2)	(3)	(4)	(5)	(6)
Variable	*hq*	*co*2	*hq*	*hq*	*co*2	*hq*
*fc*	−0.2066 ***	−6.8262 ***	−0.1552 ***	−0.4761 ***	−1.9145 **	−0.4725 ***
	(0.0360)	(0.6436)	(0.0364)	(0.0583)	(0.7937)	(0.0583)
*co*2			0.0075 ***			0.0019
			(0.0010)			(0.0020)
*gdp*	−0.2311 ***	−2.0095 **	−0.2160 ***	−0.1472 *	10.8209 ***	−0.1674 **
	(0.0344)	(0.6152)	(0.0342)	(0.0809)	(1.2652)	(0.0832)
*fdi*	2.8507 ***	22.7341 **	2.6794 ***	0.7007	−47.3798 **	0.7894
	(0.6142)	(10.9705)	(0.6091)	(0.8038)	(17.8602)	(0.8117)
*sec*	0.0070 ***	0.0291	0.0068 ***	0.0199 ***	−0.3290 ***	0.0205 ***
	(0.0015)	(0.0271)	(0.0015)	(0.0028)	(0.0478)	(0.0030)
*infra*	0.0027 ***	−0.0445 ***	0.0030 ***	0.0059 ***	0.0167	0.0059 ***
	(0.0007)	(0.0131)	(0.0007)	(0.0011)	(0.0144)	(0.0011)
*popu*	0.0400 **	−0.1696	0.0413 **	−0.6053 **	16.4634 **	−0.6362 **
	(0.0182)	(0.3255)	(0.0181)	(0.2973)	(7.8915)	(0.2932)
*infor*	−0.9138	35.7221 **	−1.1829	−1.6706	15.0590	−1.6988
	(0.8607)	(15.3728)	(0.8537)	(1.1349)	(9.4413)	(1.1362)
*cap*	−0.1386 ***	13.3845 ***	−0.2394 ***	−0.3603 ***	0.2813	−0.3609 ***
	(0.0208)	(0.3718)	(0.0247)	(0.0525)	(0.7348)	(0.0527)
*_cons*	5.7464 ***	−156.0570 ***	6.9220 ***	5.6976 ***	−13.6841	5.7232 ***
	(0.2512)	(4.4869)	(0.2951)	(1.1239)	(31.0651)	(1.1158)
*year*	NO	NO	NO	YES	YES	YES
*city*	NO	NO	NO	YES	YES	YES
*N*	2993	2993	2993	2989	2989	2989
*R−sq*	0.127	0.580	0.143	0.221	0.942	0.221

Significant at * 10%, ** 5%, and *** 1%.

**Table 5 ijerph-19-02386-t005:** Heterogeneous impact of different quantile financing constraints on high-quality urban development.

Model	(1)	(2)	(3)	(4)	(5)
Content	Above Q10	Above Q25	Above Q50	Above Q75	Above Q90
Variable	*hq*	*hq*	*hq*	*hq*	*hq*
*fc*	−0.4530 ***	−0.4714 ***	−0.5511 ***	−0.5936 **	−0.6986
	(0.0586)	(0.0665)	(0.0981)	(0.1834)	(0.4345)
*gdp*	−0.0868	0.0320	0.2183	0.4574 *	1.3778 **
	(0.0843)	(0.1032)	(0.1572)	(0.2591)	(0.4381)
*fdi*	1.2248	1.4842	1.6551	1.1656	−2.0634
	(0.8772)	(1.1260)	(1.8049)	(3.2049)	(8.5657)
*sec*	0.0180 ***	0.0165 ***	0.0086 *	0.0032	0.0053
	(0.0031)	(0.0037)	(0.0051)	(0.0092)	(0.0176)
*infra*	0.0051 ***	0.0051 **	0.0064 **	0.0039	0.0032
	(0.0012)	(0.0016)	(0.0025)	(0.0034)	(0.0042)
*popu*	−0.2423	−0.3974	−0.2802	−0.2179	−2.8257 *
	(0.3458)	(0.3961)	(0.5993)	(0.9718)	(1.4845)
*infor*	−1.8307	−2.5235 *	−3.8291 **	−4.1727 *	−3.5177
	(1.2442)	(1.2872)	(1.5741)	(2.5081)	(4.5081)
*cap*	−0.3933 ***	−0.4492 ***	−0.4891 ***	−0.5349 **	−0.8732 **
	(0.0556)	(0.0674)	(0.1045)	(0.1672)	(0.2736)
*_cons*	6.9083 ***	6.1454 ***	5.9085 **	4.8974	−7.9714
	(1.3227)	(1.5329)	(2.3257)	(3.8367)	(6.1503)
*year*	YES	YES	YES	YES	YES
*city*	YES	YES	YES	YES	YES
*N*	2715	2309	1643	875	339
*R−sq*	0.211	0.201	0.166	0.146	0.194

Significant at * 10%, ** 5%, and *** 1%.

**Table 6 ijerph-19-02386-t006:** Heterogeneous impact of financing constraints at different stages on high-quality urban development.

Model	(1)	(2)	(3)	(4)
Content	Before 2006	Before 2010	Before 2014	Before 2017
Variable	*hq*	*hq*	*hq*	*hq*
*fc*	−0.3357 **	−0.7370 ***	−0.7263 ***	−0.4761 ***
	(0.1554)	(0.0907)	(0.0668)	(0.0583)
*gdp*	0.7758 ***	−1.1014 ***	0.0398	−0.1472 *
	(0.1689)	(0.1414)	(0.0919)	(0.0809)
*fdi*	−3.6527 **	−0.2396	0.7133	0.7007
	(1.2678)	(1.2370)	(0.9078)	(0.8038)
*sec*	0.0167 **	−0.0020	0.0130 ***	0.0199 ***
	(0.0053)	(0.0051)	(0.0035)	(0.0028)
*infra*	0.0037	0.0011	0.0028 **	0.0059 ***
	(0.0081)	(0.0025)	(0.0012)	(0.0011)
*popu*	1.3183 **	−1.7745 **	−0.8033 **	−0.6053 **
	(0.4371)	(0.6670)	(0.3650)	(0.2973)
*infor*	6.9119 ***	1.9234	−1.3614	−1.6706
	(1.4454)	(1.2649)	(1.1890)	(1.1349)
*cap*	−0.1042	0.1241	−0.3512 ***	−0.3603 ***
	(0.1046)	(0.0821)	(0.0597)	(0.0525)
*_cons*	0.3496	4.9031 *	3.7068 **	5.6976 ***
	(1.9784)	(2.6238)	(1.3814)	(1.1239)
*year*	YES	YES	YES	YES
*city*	YES	YES	YES	YES
*N*	810	1898	2722	2989
*R−sq*	0.336	0.372	0.163	0.221

Significant at * 10%, ** 5%, and *** 1%.

## Data Availability

Data are available on request from the authors.

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
