# Peer review of "Financing Constraints, Carbon Emissions and High-Quality Urban Development—Empirical Evidence from 290 Cities in China"

_ijerph, 2022, doi:10.3390/ijerph19042386_

Round 1

Reviewer 1 Report

The topic presented in this work is really interesting. However, several challenges are required:

I analyze the single sections:

Abstract has inappropriate structure. I suggest to answer the following aspects: - general context - novelty of the work - methodology used (describe briefly the main methods or treatments applied) - main results and related interpretations.

Introduction: This section should briefly place the study in a wide context and emphasize why it is relevant carrying out the analysis. It should define the purpose of the work and its significance. In this perspective, this section is too succinct and fails to effectively point out the relevance of your contribution towards the existing literature. Moreover, the authors do not provide at the end of the section the description of the paper structure which is very useful for readers.

Literature section is well done. However, I would strongly suggest the authors to look also at the role of green finance and green investments on sustainability of the environments.

https://www.sciencedirect.com/science/article/pii/S0959652620318461

https://www.inderscienceonline.com/doi/abs/10.1504/IJGE.2020.109735

Materials and methods: I found this section very important for the readability of the paper. The research methodology seems underdeveloped. Methods should be described in detail. I think the research procedure could be much more clearly described by means of a diagram also highlighting its potential and limit.

Discussions: The discussion of the results is merely descriptive and the obtained evidence is flimsy due to the fact the outcomes are not supported by an adequate discussion in light of scientific literature. Authors should discuss the results and how they can be interpreted in perspective of previous studies and their implications should be discussed in the broadest context possible.

Conclusions must also be revised according to the previous comments. In particular, they should discuss practical and policy implications as well as future lines of research. As it stands now, they fail to extract all the juice of your work. 

I hope these comments might help in improving the paper and encourage the authors to move forward.

Reviewer 2 Report

The study addresses the highly relevant issue of mitigating carbon pollution and improving the quality of urban development in the case of China. However, I would suggest the authors restructure the manuscript to improve the readability and contribution to the literature. 

In the Abstract, consider briefly outlining the overall relevance of the topic and then proceed with the specific relevance for China. Explain the gaps that exist and how the study attempts to address them. Describe the methodological approaches used and summarize findings. Finally, explain the outcomes and implications - both practical policies and theoretical research directions. The abstract must be precise and focused.

I consider sections 1-3 ineffective and recommend restructuring. In the Introduction, the author should expand the abstract pattern (see above). Detail the relevance of the study not only for China, but globally, and then proceed with China. The author should clearly explain the relevance, identify problems and challenges, and articulate the goal of the study based on a comprehensive discussion of these problems and challenges. 

Contributions and findings should be summarized in the conclusion, not the Introduction.

In the Introduction, Support your reasoning by citing relevant literature. Currently, the literature review does not result in any summary of problems and gaps. It leads nowhere, it is just a segmented overview of current literature related to the topic, nothing more. The authors should summarize their literature review by identifying problems (see above) they attempt to solve in their study. This should be done in the Introduction, this narrative should be integrated with the discussion of the relevance of the topic. From this discussion, the authors should formulate their goals.

The Discussion section must address previous studies - currently, there are none. Authors must compare their findings with previous studies, demonstrate agreements and disagreements, and discuss their findings critically. The contribution of the study to the literature must be clearly demonstrated and emphasized. Novelties (compared to other studies) should be summarized, potential conflicts with previous findings should be articulated, further research directions should be outlined. 

In the Conclusion, the authors should address limitations and problems they have faced when building and implementing the model and suggest potential solutions to overcoming these limitations. 

Reviewer 3 Report

Review of the paper: “Financing Constraints, Carbon Emissions and High-Quality Urban Development—Empirical Evidence from 290 Cities in  China”. I find the title formulated in this way misleading. The reservation concerns the statement "High-Quality Urban Development". This is not High-Quality Urban Development, but only a model of a city with high dynamics of industrial development. From this perspective, reductions in CO2 emissions actually reduce the development opportunities of certain types of industry. The presented approach corresponds to a historical approach, for example, the cities of the global North in the 18th and 19th centuries were focused on maximizing industrial production, but were not designed to minimize health and environmental risks for their inhabitants. As a result, they were not a healthy or even comfortable place to live. The transformation of these cities was based on a deliberate redesign to ensure the safety of residents. However, such an urban economy model is largely dominated by solutions designed to isolate or remove threats. In this context, and in the face of new development conditions facing cities in the planning practice in the second half of the 20th century, new concepts of urban development related to the trend of sustainable development appeared.
In view of the above, I cannot agree with the assumptions regarding the definition of “High-Quality Urban Development” and the use of TFP for this purpose, where the essence of high-quality urban development is production efficiency and development potential. I believe that contemporary development should be viewed through the prism of many perspectives (this approach is mentioned in paragraphs 206-207). Such an approach completely ignores the quality of life in the city and ecological aspects, which in turn leads to conclusions inconsistent with current knowledge. For example, as an economist who deals with environmental problems, it is difficult for me to accept such statements as, for example:

•    paragraphs 402 – 403 “Therefore, the current increase in China’s carbon emissions is necessary for the improvement of high-quality urban development”.
•     paragraphs 442 – 443 “The visible countermeasures should reduce financing constraints and increase carbon emissions to promote high-quality urban development.”
That's why, I suggest removing the adjective "High-Quality Urban Development" as confusing. It can be replaced, for example, by “Industrial Development”, which changes the reception of the article.
Kind regards,

Round 2

Reviewer 1 Report

The manuscript is improved

Reviewer 2 Report

My recommendations have been addressed

Reviewer 3 Report

Dear Authors,

I accept  the response to the review, and also I find the changes you made to the text of the article as satisfactory.

Kind regards,